# Antidiabetic, Antihyperlipidemic, and Antioxidant Evaluation of Phytosteroids from *Notholirion thomsonianum* (Royle) Stapf

**DOI:** 10.3390/plants12203591

**Published:** 2023-10-17

**Authors:** Mohammad A. Huneif, Shah Fahad, Alqahtani Abdulwahab, Seham M. Alqahtani, Mater H. Mahnashi, Asif Nawaz, Fida Hussain, Abdul Sadiq

**Affiliations:** 1Pediatric Department, Medical College, Najran University, Najran 61441, Saudi Arabia; maalhuneif@nu.edu.sa (M.A.H.); aaalsharih@nu.edu.sa (A.A.); drseham2015@gmail.com (S.M.A.); 2Department of Agronomy, Abdul Wali Khan University Mardan, Mardan 23200, KP, Pakistan; shahfahad@awkum.edu.pk; 3Department of Pharmaceutical Chemistry, College of Pharmacy, Najran University, Najran 61441, Saudi Arabia; 4Department of Pharmacy, Faculty of Biological Sciences, University of Malakand, Chakdara 18000, KP, Pakistan; asifnawaz2446@gmail.com; 5Department of Pharmacy, University of Swabi, Swabi 23561, KP, Pakistan; fida2k9@yahoo.com

**Keywords:** *Notholirion thomsonianum*, diabetes, in vivo and in vitro studies, phytosteroids, blood biochemistry

## Abstract

Diabetes mellitus (DM) is a metabolic complication and can pose a serious challenge to human health. DM is the main cause of many life-threatening diseases. Researchers of natural products have been continuously engaged in treating vital diseases in an economical and efficient way. In this research, we extensively used phytosteroids from *Notholirion thomsonianum* (Royle) Stapf for the treatment of DM. The structures of phytosteroids **NtSt01** and **NtSt02** were confirmed with gas chromatography–mass spectrometry (GC-MS) and nuclear magnetic resonance (NMR) analyses. Through in vitro studies including α-glucosidase, α-amylase, and DPPH assays, compound **NtSt01** was found to be comparatively potent. An elevated dose of compound **NtSt01** was also found to be safe in an experimental study on rats. With a dose of 1.0 mg/kg of **NtSt01**, the effect on blood glucose levels in rats was observed to be 519 ± 3.98, 413 ± 1.87, 325 ± 1.62, 219 ± 2.87, and 116 ± 1.33 mg/dL on the 1st, 7th, 14th, 21st, and 28th, days, respectively. The in vivo results were compared with those of glibenclamide, which reduced the blood glucose level to 107 ± 2.33 mg/dL on the 28th day. On the 28th day of **NtSt01** administration, the average weights of the rats and vital organs (liver, kidney, pancreas, and heart) remained healthy, with a slight increase. The biochemical parameters of the blood, i.e., serum creatinine, blood urea, serum bilirubin, SGPT (or ALT), and serum alkaline phosphatase, of rats treated with **NtSt01** remained in the normal ranges. Similarly, the serum cholesterol, triglycerides, high-density lipoprotein (HDL), and low-density lipoprotein (LDL) levels also remained within the standard ranges. It is obvious from our overall results that the phytosteroids (specifically **NtSt01**) had an efficient therapeutic effect on the blood glucose level, protection of vital organs, and blood biochemistry.

## 1. Introduction

As a chronic disease, DM is a metabolic disorder in which insulin production by the pancreas is reduced or the produced insulin may be ineffective, leading to hyperglycemia, which can cause further damage to other body systems like the circulatory and nervous systems. Some of the abnormalities that account for DM include defects in the production of insulin or in its action or secretion and dysfunction in the metabolism of fat, protein, and carbohydrates [1,2,3]. Polyphagia, polyuria, and polydipsia are the some of the symptoms associated with the state of hyperglycemia [4]. There are about 450 million diabetic patients around the globe, with this figure expected to reach 690 million by the year 2044 [5]. Currently, DM prevalence has been reported as 8.5% of adults globally, and a rapid increase has been observed in countries with a low or middle income [6]. DM is a serious metabolic disease, and chronic hyperglycemia can cause numerous complications and the dysfunction of several organs like the kidneys, nerves, eyes, heart, blood vessels, and liver [2,7,8].

There are different types of DM; the common types are type 1 and type 2 DM. Type 1 DM is insulin-dependent and is due to insulin deficiency, along with an impairment of the β cells of the pancreas [9], whereas type 2 DM is non-insulin-dependent and is due to insulin resistance or decreased insulin secretion [10]: 95% of diabetic patients have type 2 DM [11]. DM development in people with an impaired tolerance of glucose can be prevented or managed with the use of antidiabetic drugs or through changing their lifestyle via exercise, diet control, and/or weight loss [12]. The use of antidiabetic drugs can have number of adverse effects, including hypoglycemia, retention of fluids, osteoporosis, and heart failure, due to which their use is limited [13,14,15]. Hence, the development of new, effective drugs that have fewer adverse effects is needed to control and manage diabetes. In the development of antidiabetic drugs that are specific for type 2 DM, several biochemical approaches can be used. Among the important biochemical pathways, the inhibition of α-amylase and α-glucosidase is common. Both of these enzymes break down starch and oligosaccharides into glucose, leading to an increase in the concentration of glucose; hence, their inhibition is important for decreasing glucose absorption in the intestine [16].

For thousands of years, medicinal plants and natural products have been reported for the treatment of many diseases, including diabetes, especially type 2 DM [17,18,19]. The crude extracts of medicinal plants and their bioactive compounds have been found to be useful in many pharmacological activities [20,21,22]. Approximately 400 plants have been demonstrated to have antidiabetic activity, but only some of them have been evaluated for their efficacy [23]. A number of natural products of plant origin have been shown to have an antidiabetic activity. The most important reported phytochemicals include alkaloids, carbohydrates, peptidoglycan, amino acids, glycosides, steroids, glycopeptides, galactomannan gum, terpenoids, hypoglycans, guanidine, and inorganic ions [24]. Different plants and microorganisms produce α-glucosidase and α-amylase inhibitors for the regulation of such enzyme activities [25]. Synthetic compounds are also being developed in parallel in order to create α-glucosidase and α-amylase inhibitors [26,27]. Inhibitors of α-amylase enzymes reduce the conversion of starch into glucose usually after eating a meal, which results in a decrease in the level of glucose in the blood. Therefore, α-amylase inhibitors are needed to control the glucose levels of diabetic patients.

*Notholirion thomsonianum* is a small bulbous plant of the family Liliaceae. This small lilium-like medicinal plant has been studied for various pharmacological effects [28]. The plant can be used to improve the digestive system and in the management of microbial infections [29]. Our group has been exploring the medicinal aspects of this species for a decade. We have previously explored its crude extract for antibacterial, antifungal, and analgesic effects [29,30]. In recent years, we explored the potential of this plant for the management of diabetes mellitus using its various fractions and some bio-guided bioactive compounds following multitarget in vitro and in silico approaches [31]. Based on our previous experience with this plant, its hydroalcoholic extract contains phytosteroids, which have been identified. This current study is extensive compared with our previously published work. In this study, we extensively used these phytosteroids for investigating in vitro and in vivo antidiabetic targets. Furthermore, the beneficial effects of the identified phytosteroids were also explored on the vital organs of the body like the liver, kidney, pancreas, and heart and on blood biochemistry.

## 2. Results

### 2.1. Phytochemistry

In this research, we initially purified and identified two different phytosteroids (**NtSt01** and **NtSt02**, as shown in Figure 1). The isolated amount of **NtSt01** was 830 mg as a white powder, while 375 mg of **NtSt02** was isolated as a yellowish-brown solid. The structures of these two isolated compounds were initially confirmed with GCMS analysis. The retention time of compound **NtSt01** was 56.964 min, with a base peak value of 55.1 (Appendix A). The fragmentation pattern of compound **NtSt01** is shown in the Appendix A. The spectrum and its fragmentation pattern were compared with the library spectrum and the difference spectrum, as shown in Appendix A, respectively. The chemical name of the identified compound **NtSt01** is 3-β-Acetoxystigmasta-4,6,22-triene with an IUPAC name of (E)-17-(5-ethyl-6-methylhept-3-en-2-yl)-10,13-dimethyl-2,3,8,9,10,11,12,13,14,15,16,17-dodecahydro-1H-cyclopenta[a]phenanthren-3-yl acetate. Similarly, from the same GCMS analysis, the compound **NtSt02** was observed at a retention time of 57.812 min, with a major peak at *m*/*z* 135 (Appendix A). The fragmentation pattern of the compound **NtSt02** is shown in Appendix A. This spectrum and its fragmentation pattern were compared with the library spectrum and difference spectrum, as shown in Appendix A, respectively. The chemical name of the identified compound **NtSt02** is 4,6-cholestadien-3beta-ol, benzoate, and the IUPAC name is 10,13-dimethyl-17-(6-methylheptan-2-yl)-2,3,8,9,10,11,12,13,14,15,16,17-dodecahydro-1H-cyclopenta[a]phenanthren-3-yl benzoate.

### 2.2. Alpha Glucosidase Inhibition

The in vitro α-glucosidase inhibitory results of both phytosteroids of *N. thomsonianum* (**NtSt01** and **NtSt02**) are shown in Table 1. The percent inhibitions were recorded on all concentrations in triplicate. **NtSt01** was three times more potent than **NtSt02**, as observed from their respective IC_50_ values. **NtSt01** exhibited inhibitions of 85.00 ± 1.52, 81.52 ± 1.85, 77.63 ± 1.56, 68.78 ± 1.02, and 61.22 ± 0.85% at experimental concentrations of 500, 250, 125, 62.50, and 31.25 μg/mL, respectively. The IC_50_ values of **NtSt01** and **NtSt02** were 7.34 and 22.87 μg/mL, respectively, in comparison to the standard drug, acarbose, with an IC_50_ value of 2.14 μg/mL.

### 2.3. Alpha Amylase Inhibition

The in vitro α-amylase activities of **NtSt01** and **NtSt02** were also analyzed in comparison to the standard acarbose, as shown in Table 2. Likewise, **NtSt01** was found with very practical result which was eleven folds more potent than **NtSt02**. The IC_50_ value of **NtSt01** and **NtSt02** were 4.17 and 46.73 μg/mL respectively in comparison to the standard drug acarbose with the IC_50_ value of 1.96 μg/mL. The **NtSt01** demonstrated percent inhibitions of 80.03 ± 2.11, 75.52 ± 0.96, 71.63 ± 0.92, 67.63 ± 2.51 and 62.35 ± 1.78% at experimental concentrations of 500, 250, 125, 62.50 and 31.25 μg/mL respectively.

### 2.4. Antioxidant Assay

Antioxidant activity is a very important supplementary activity in antidiabetic studies. Antioxidants combat excessive free radicals within the body. This concept applies to the reduction of free radicals in the pancreas, which partially protects the pancreas from damage and helps with diabetic control. Keeping this concept in mind, we also determined the antioxidant activity of the phytosteroids. Though both of our phytosteroids demonstrated low antioxidant activity profiles, they were both notable as supplementary targets. The observed IC_50_ values for **NtSt01** and **NtSt02** were 142.76 and 223.43 μg/mL, respectively, as shown in Table 3.

### 2.5. In Vivo Results

Based on the enzymatic assays, we observed that **NtSt01** is a potential inhibitor of α-glucosidase and α-amylase. With these results, we further analyzed the compound **NtSt01** in in vivo studies using experimental rats following ethical guidelines. The dose of **NtSt01** was started at 200 and was increased up to 2000 mg/kg body weight. The rats remained healthy; no morbidity, mortality or irritation was observed for the tested doses during the acute toxicity studies.

Phytosteroid **NtSt01** (1.0 mg/kg) was administered to alloxan-induced fasting diabetic rats, and the blood glucose levels were observed until 28 days, as shown in Table 4. The blood glucose level in the normal control group remained within the normal range during the observational time. In contrast, the blood glucose level of the diabetic control group remained elevated throughout the experiment. At 1.0 mg/kg of **NtSt01**, the effect on the blood glucose levels in rats were 519 ± 3.98, 413 ± 1.87, 325 ± 1.62, 219 ± 2.87 and 116 ± 1.33 mg/dL on days 1, 7, 14, 21, and 28, respectively. In comparison, in the glibenclamide control group (0.5 mg/kg), the blood glucose level dropped from 517 ± 0.77 (day 1) to 103 ± 1.70 mg/dL (day 28).

During the observational time period, the effect on the body weight (in grams) of the rats was also closely observed (as shown in Table 5). In the normal control group, the body weight of the rats remained the same throughout the four weeks. In the diabetic control group, we observed a body weight loss of 18 g of rats at four weeks. With the administration of phytosteroid **NtSt01** at a concentration of 1.0 mg/kg, the rats became healthy, and there was a gain in body weight. The body weight changed from 238 ± 0.33 g (day 1) to 246 ± 0.11 g (day 28) in our sample. In comparison, the weight of the rats in the glibenclamide control group also increased slightly.

Diabetes mellitus is a perilous disease, and it affect all the vital organs of the body. After the in vivo experiments, the rats from the different groups were euthanized following the ethical procedure. The vital organs were isolated, and the weights were recorded, as shown in Table 6. The weights of the liver, kidney, pancreas, and heart were 9.83 ± 0.92, 1.01 ± 0.15, 0.92 ± 0.12, and 1.04 ± 0.13 g, respectively, in the **NtSt01**-administered group. In diabetic control group, we observed a drastic effect on the average weights of the vital organs.

The results of the renal function and liver function tests are summarized in Table 7. The serum creatinine, blood urea, serum bilirubin, ALT, and serum alkaline phosphatase of the normal, diabetic, and standard (glibenclamide) groups were compared with the results of phytosteroid **NtSt01**. In the normal group, all the parameters were within the normal range. In the diabetic group, elevated blood urea, ALT, and serum alkaline phosphatase levels were observed. In a pattern like that of the standard group, the phytosteroid **NtSt01** group exhibited normal serum creatinine, blood urea, and serum bilirubin values. However, the ALT and serum alkaline phosphatase values were near the upper normal limit. These overall results showed that compound **NtSt01** was effective in protecting the vital organs from damage over the four weeks of the experiment.

The lipid profile is a major concern in the evaluation of serum triglycerides, total cholesterol, high-density cholesterol (HDL), and low-density cholesterol (LDL). The results of lipid profile of all the experimental groups are summarized in Table 8. Except for the serum cholesterol, all other values were within the normal ranges.

### 2.6. Molecular Docking Studies

For the determination of the pharmacological parameters of the ligand with targeted protein moieties, using the fit model theory with ligand–enzyme interactions, docking studies were performed. These docking results were examined to understand the different interaction parameters. **NtSt01** was docked with the targeted macromolecules to analyze the binding affinity. It was found to have excellent binding affinity with an energy of −8.84 Kcal/mol against α-glucosidase 7K9Q and -8.9 Kcal/mol against α-amylase 4W93. The results of the interaction with α-glucosidase are depicted in Figure 2. The prominent interactions are Arg 500 and Asn 551, respectively.

**NtSt01** also displayed excellent results, with great binding affinities against α-amylase. The results are depicted in Figure 3. The prominent interactions were found to be with Trp 59, His 101, Leu 162, His 201, Ala 198, and Ile 235, respectively.

## 3. Materials and Methods

### 3.1. Phytochemistry

We collected rhizomes of *Notholirion thomsonianum*, which were also used in our previous research [29,30,31]. In this study, we subjected the hydroalcoholic extract of *N. thomsonianum* to isolation. Initially, we used a large-diameter gravity column to partially purify the sample. The solvent system used in this chromatography included n-hexane and ethyl acetate. Initially, the column was started with pure n-hexane for a 100 mL elution. Then, the polarity of solvent was gradually increased by adding 5% ethyl acetate each time, and the fractions were collected. The fractions were preliminary checked via thin-layer chromatography (TLC) analysis. One of the fractions containing two major and a few minor spots was concentrated and subjected to a small column for further purification. The elution fractions were monitored closely and collected in separated vials. The vials with major coeluted spots were combined and dried via a rotary evaporator. The two major spots were subjected to GCMS analysis for the identification of the components as per the previously described procedure [32].

### 3.2. Alpha-Glucosidase Inhibition

The alpha-glucosidase inhibitory effects of our compounds were determined via a reported protocol [31]. Sample dilutions (50 µL) and 100 µL of α-glucosidase solution (0.5 U/mL) were mixed. Then, 600 μL of phosphate buffer (0.1 M, pH 6.9) was added, and the mixture was incubated at 37 °C for 15 min. After this, 5 mM substrate (p-nitrophenyl α-D-glucopyranoside) solution (20 µL) in 0.1 M phosphate buffer (pH 6.9) was added and again incubated under the same conditions. Sodium carbonate was added to stop the reaction; at 405 nm, the absorbance was noted using a spectrophotometer. A mixture without α-glucosidase served as the blank, and a mixture without the test compound served as the control. The percent enzyme inhibition was calculated as:(1)$$\%\;Alpha\;glucosidase\;inhibition=\frac{Control\;abs.-Sample\;abs.}{Control\;abs.}\times 100$$

### 3.3. Alpha-Amylase Inhibition

The inhibitory activity of our compounds against alpha-amylase enzymes was determined as per the previously reported protocols [31]. Test samples of 100 µL were mixed with an enzyme solution (200 µL) and 100 µL of phosphate buffer (pH 6.9, 2 mM). The mixture was then incubated at 25 °C for 20 min, followed by the addition of 100 µL of starch solution (1%). The same procedure was followed for positive controls, where phosphate buffer was added instead of enzyme solution (200 µL). After 5 min of incubation, 3,5-dinitrosalicylic acid reagent (500 µL) was added to the test samples and control group. The mixtures were incubated again for 10 min; at 580 nm, the absorbance was recorded using a spectrophotometer. The alpha-amylase percent inhibition was calculated as:(2)$$\%\;Amylase\;inhibition=\left[1-\left(\frac{Test\;sample\;abs.}{Control\;abs.}\right)\right]\times 100$$

### 3.4. Antioxidant Assay

To evaluate the antioxidant potential of our compounds, the DPPH free radical scavenging assay developed by Brand William was used [33]. The DPPH solution was made by dissolving 20 mg of DPPH in 100 mL of methanol, and its absorbance at 517 nm was adjusted to 0.75. Then, 2 mL of DPPH solution was added to 2 mL of sample dilutions ranging from 31.25 to 500 µg/mL, and the mixture was incubated at room temperature in the dark for 15 min. The absorbance was noted at 517 nm, and the following formula was used to determine the % inhibition of DPPH free radicals.
(3)$$\%\;Inhibition=\frac{Control\;abs.-Sample\;abs.}{Control\;abs.}\times 100$$

### 3.5. Molecular Docking Studies

The docking studies were performed using Auto Dock Vina 1.2.2. PyRx. The three-dimensional structure of ligand **NtSt01** was drawn in ChemDraw 20.0 software and saved as a Mol.file. The structure was modified by adding polar hydrogen using Discovery Studio Visualizer and saved in PDB format. The three-dimensional structures of both targeted proteins, α-glucosidase and α-amylase, were acquired from the RCSB protein data bank (http://www.rcsb.org accessed on 22 September 2023) as PDB id 4W93 and 7K9Q, respectively, and saved in PDB format. Before starting the computational studies on the ligand, the docking process protocols were validated through a redocking process. These ligands and targeted protein structures were permitted for energy minimization through the Charm force field factor, which detached the unwanted crystallographic observations. The ligand and targeted protein structures were opened in Autodock Vina and converted to ligands and macromolecules as Pdbqt molecules. The grid box was adjusted as center X: 10.112, Y: 68.356, and Z: 32.853, with dimensions (Angstrom) X: 75.2435, Y: 105.3254, and Z: 103.4758. The results were visualized through Discovery Studio Visualizer software 2017 R2 [22].

### 3.6. In Vivo Experiments

#### 3.6.1. Experimental Animals and Ethical Approval

The experimental rats were purchased from the Breeding House of National Institute Health, Islamabad, Pakistan. The average weights of these rats ranged from 220 to 250 g. The animals were handled as per the guidelines of the University of Malakand animals By-Laws 2008 (Scientific Procedure Issue I) under the approval of the ethical committee via letter No. DREC-140/B. The food, water, and light/dark cycles provided to animals were observed by the ethical committee [34]. The animals were euthanized as per the AVMA guidelines version 2020. The animals were subjected to slow exposure of halothane vapors to induce anesthesia. There was a gradual increase in the dosage of halothane vapors, which eventually euthanized the animals [35].

#### 3.6.2. Acute Toxicity

In experimental animals, acute toxicity studies were performed to determine the safe dose of our test compounds for in vivo studies. Test compounds were administered at increasing doses of up to 2000 mg/kg, and rats were observed for any aberrant behavior or lethality [34].

#### 3.6.3. Induction of Diabetes

Alloxan monohydrate (Sigma Aldrich; Steinhein, Switzerland) was used to induce diabetes at a dose of 160 mg/kg in the experimental animals [36]. The experimental animals were kept in fasting mode for 8–12 h but allowed water only before being subjected to the bioassay. The level of blood glucose was checked after 48 h of administration of alloxan using a glucometer, and only the animals that were diabetic were considered for the study, having a blood glucose level of more than 200 mg/dL [34].

#### 3.6.4. Experimental Design

The antidiabetic activity of our test compounds was studied in alloxan-induced diabetic animals. All the experimental animals were placed into four different groups, with 6 animals in each group.

**Group 1:** Only normal saline i.p was given to this nondiabetic group throughout the experimental period.

**Group 2:** Alloxan was administered i.p to this group for diabetes induction and rats were under observation without any treatment throughout the experiment.

**Group 3:** This group was the alloxan-induced diabetic group, which was treated with 0.5 mg/kg of the standard drug, glibenclamide.

**Group 4:** This group was the alloxan-induced diabetic group, which was treated with 1.0 mg/kg of **NtSt01** intraperitoneally.

#### 3.6.5. Lipid Profile

Standard methods were used for the analysis of total cholesterol, serum triglycerides, low-density lipoprotein (LDL), and high-density lipoprotein (HDL). Briefly, 10 mL of serum sample was added to 1000 mL solution of triglyceride. The mixture was incubated for 10 min at 37 °C and the absorbance was recorded at 546 nm. To monitor blood cholesterol, diagnostic kits were utilized following the specifications of the manufacturer. A total of 10 µL of serum sample was added to 1000 µL of cholesterol solution, followed by incubation for 10 min at 37 °C; the absorbance against blank was noted at 546 nm. For HDL concentration determination, 200 µL of serum sample and 500 µL of HDL solution were combined, and the mixture was allowed to stand at room temperature for 5 min. Again, the mixture was mixed and subjected to further centrifugation for 5 min, followed by supernatant collection. Then, 50 µL of supernatant of HDL was added to 500 µL of solution of cholesterol; the mixture was incubated at 37 °C for 5 min. Then, at 546 nm, the sample absorbance was measured. Similarly, LDL was determined in all groups following the manufacturer’s specifications [37].
LDL = Total cholesterol + HDL − Triglycerides/5(4)

#### 3.6.6. Renal Functions Tests

Renal functions tests were performed to determine blood urea and serum creatinine [38]. To determine blood urea, enzyme reagent 1 (1000 µL) and 10 µL of serum were mixed together and incubated for 5 min at 25 °C, followed by the addition of 1000 µL of reagent 2 to the mixture. After 5 min, the absorbance was recorded at 578 nm. The serum creatinine was measured very carefully, as it is highly sensitive reaction to temperature. Then, 500 µL of reagent and 50 µL of serum were mixed, followed by incubation for 1 min at 37 °C, and the absorbance was noted at 500 nm.

#### 3.6.7. Liver Functions Tests

Serum glutamate pyruvate transaminase (SGPT/ALT), alkaline phosphatase (ALP), and bilirubin tests were performed following standard protocols using micro lab 300 and tecno plus biochemistry analyzers [39]. To perform the SGPT/ALT test, 50 µL of serum sample was added to 500 µL of reagent (400 µL of reagent 1 and 100 µL of reagent 2). The mixture was incubated at 37 °C for 30 s, and the absorbance was noted at 340 nm. To determine ALP, a kit was used following the manufacturer’s specifications: 10 µL of serum sample was added to 500 µL of reagent (400 µL of reagent 1 and 100 µL of reagent 2). The mixture was then incubated at 37 °C for 30 s, and the absorbance was noted at 405 nm. Similarly, the concentration of bilirubin was evaluated using standard protocols. Four types of reagents, R1, R2, R3, and R4, were used in this test. R2 (25 µL) was added to R1 (100 µL). After this, 100 µL of serum sample and 500 µL of R3 were added and allowed to stand at 25 °C for 5 min. Then, 500 µL of R4 was added; at 25 °C for 5 min, the mixture was incubated. The absorbance of the sample was recorded at 546 nm.

#### 3.6.8. Statistical Analysis

Two-way ANOVA followed by Dunnett’s post-test were applied for the comparison of the positive control with the test groups using GraphPad prism 8.0.1 software. *p* values less than or equal to 0.05 were considered statistically significant. The findings of the statistical analysis are shown as mean ± SEM.

## 4. Discussion

For thousands of years, number of medicinal plants and natural products have been reported for the treatment of many diseases including diabetes [40,41,42]. Metformin, obtained from *Galega officinalis*, has been the first-line drug used for 60 years for the treatment of type 2 diabetes [43]. Plant-based drugs have minimal or no side effects compared with synthetic drugs; therefore, researchers have mainly focused on natural products for developing new drugs. In DM, the main treatment strategy involves controlling high glucose levels in the blood. Apart from this, α-amylase and α-glucosidase enzymes inhibitions are important strategies for controlling hyperglycemia, as these enzymes convert starch into glucose, resulting in increased levels of blood glucose [44]. For *Notholirion thomsonianum*, the in vitro antidiabetic properties of the in vitro targets of α-amylase, α-glucosidase, and tyrosine phosphatase 1B, and the antioxidant potential have been studied using free radical assays of ABTS, DPPH, and H_2_O_2_ [1]. In this study, we determined the antidiabetic potential of *Notholirion thomsonianum* through in vitro (α-amylase and α-glucosidase inhibition) and in vivo studies.

The in vitro antidiabetic properties of *Notholirion thomsonianum* were determined against α-amylase and α-glucosidase. α-Amylase converts starch into disaccharides and oligosaccharides, whereas α-glucosidase hydrolyzes disaccharides into glucose [45]. α-Amylase and α-glucosidase enzymes, if inhibited, decrease the glucose level via the breakdown of starch in GIT, which is retarded via the inhibition of these enzymes, which ameliorates hyperglycemia in diabetic patients. In the α-glucosidase inhibition assay, the compounds **NtSt01** and **NtSt02** exhibited inhibitions of 85.00 ± 1.52 and 77.56 ± 3.22% at the highest concentration (500 µg/mL), with IC_50_ values of 7.34 and 22.87 µg/mL, respectively. Compounds **NtSt01** and **NtSt02** exhibited 80.03 ± 2.11 and 74.99 ± 1.53% inhibition at 500 µg/mL against α-amylase with IC_50_ values of 4.17 and 46.73 µg/mL, respectively. However, at the same tested concentration (500 µg/mL), the standard drug acarbose demonstrated 92.65 ± 0.55% inhibition against α-glucosidase and 91.01 ± 1.36% inhibition against α-amylase with IC_50_ values of 2.14 and 1.96 µg/mL, respectively.

In the blood glucose test, diabetes induction with alloxan was confirmed, and the blood glucose levels in the diabetic group were observed as 521 ± 1.28, 517 ± 2.06, 526 ± 1.00, 524 ± 2.66, and 529 ± 2.80 mg/dL on days 1, 7, 14, 21 and 28, respectively. Treatment of diabetic animals with **NtSt01** at a dose of 1.0 mg/kg produced a decline in blood glucose levels, i.e., 519 ± 3.98, 413 ± 1.87, 325 ± 1.62, 219 ± 2.87, and 116 ± 1.33 mg/dL on the 1st, 7th, 21st, and 28th days of the treatment, respectively.

No statistical differences in the weights of the liver, kidney, pancreas, and heart were observed among all groups after completion of the experiments. The kidney function profile, lipid profile, and biochemical profile of the test animals in all groups were also determined. In the biochemical tests, blood urea, serum bilirubin, serum creatinine, SGPT(ALT), and serum alkaline phosphatase levels were high in the diabetic disease group and decreased after treatment with 1.0 mg/kg **NtSt01**, as shown in Table 7. In the lipid tests, considering serum cholesterol, serum triglycerides, LDL, and HDL, a significant decline was demonstrated in our sample (**NtSt01** 1.0 mg/kg) in comparison with the diabetic animals, as shown in Table 8. Both the in vitro and in vivo antidiabetic results for **NtSt01** revealed the potential of this compound for the management of DM and as a multitarget antidiabetic agent.

Molecular docking is an important approach to determine the binding energies and interactions of a drug molecule for its target. We docked our isolated compounds against α-glucosidase and α-amylase enzymes. The molecular docking studies revealed encouraging binding interactions with the target proteins for our compounds.

Though the whole of our work is based on the development of an antidiabetic drug from medicinal plant source, significant efforts have also been made by synthetic organic chemists for the discovery of new antidiabetic agents [46]. Among the synthetic approaches, the modification of immunosugar has been identified to develop potential inhibitors of alpha-glucosidase [47,48,49].

## 5. Conclusions

Herein, we explored the antidiabetic potential of phytosteroids from *Notholirion thomsonianum*. In an attempt to combat diabetes and its consequences, weave used phytosteroid **NtSt01** as a natural drug. In the in vitro α-glucosidase, α-amylase, and DPPH assays, compound **NtSt01** was found to be potent enough to be tested in experimental animals. The compound was initially found safe in experimental rats and was then found effective in combating induced diabetes over the four weeks of the experiment. After the experiment, we also observed that the weights of the liver, kidney, pancreases, and heart and their functions were within the allowed limits. Our overall results show that phytosteroids (specifically **NtSt01**) have an efficient therapeutic effect on the blood glucose level and other protective effects on organs like the liver, kidney, pancreases, and heart, so may serve as a multitarget antidiabetic drug.

## Figures and Tables

**Figure 1 plants-12-03591-f001:**
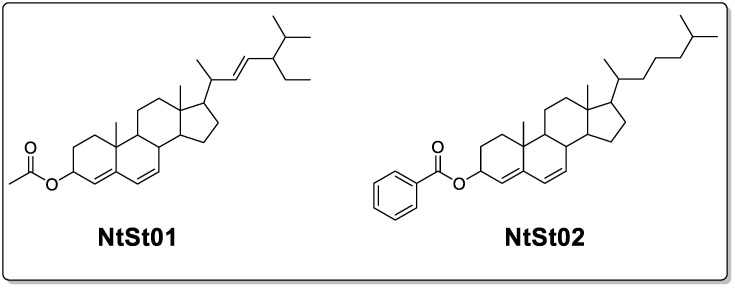
Chemical structures of the identified phytosteroids in *Notholirion thomsonianum*.

**Figure 2 plants-12-03591-f002:**
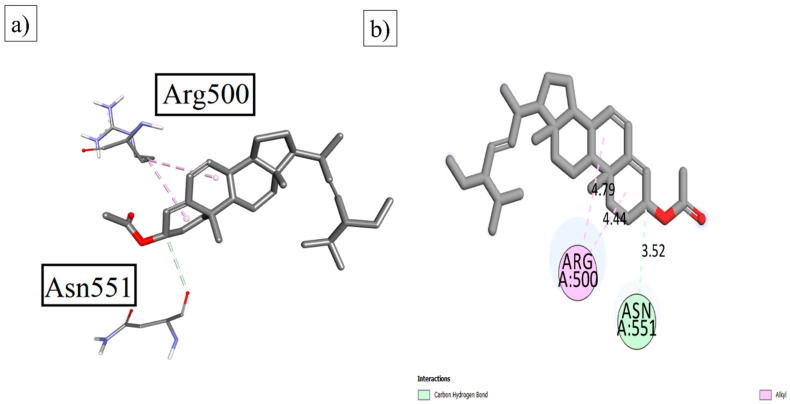
The (**a**) 3D and (**b**) 2D visualizations of **NtSt01** with α-glucosidase.

**Figure 3 plants-12-03591-f003:**
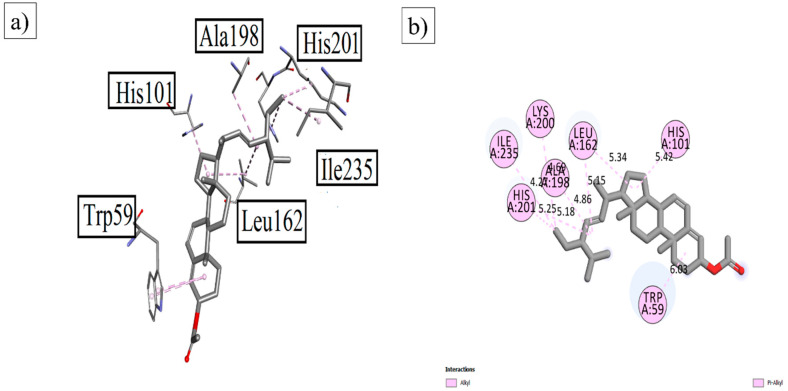
The (**a**) 3D and (**b**) 2D visualization of **NtSt01** with α-amylase.

**Table 1 plants-12-03591-t001:** Alpha-glucosidase inhibitions of the phytosteroids.

Comp/Standard	Conc (μg/mL)	Percent Inhibition (Mean ± SEM)	IC_50_ (μg/mL)
**NtSt01**	500 250 125 62.50 31.25	85.00 ± 1.52 *** 81.52 ± 1.85 *** 77.63 ± 1.56 *** 68.78 ± 1.02 *** 61.22 ± 0.85 ***	7.34
**NtSt02**	500 250 125 62.50 31.25	77.56 ± 3.22 *** 70.63 ± 2.45 *** 64.52 ± 3.15 *** 59.98 ± 1.88 *** 52.63 ± 1.52 ***	22.87
**Standard Drug**	500 250 125 62.50 31.25	92.65 ± 0.55 89.53 ± 1.45 83.89 ± 2.65 78.63 ± 1.98 70.52 ± 2.63	2.14

All the values are expressed as mean ± SEM compared with the standard. Two-way ANOVA followed by Dunnett’s test was applied. ***, significantly different (*p* < 0.001) compared with standard drug.

**Table 2 plants-12-03591-t002:** Alpha amylase inhibitions of the phytosteroids.

Comp/Standard	Conc (μg/mL)	Percent Inhibition (Mean ± SEM)	IC_50_ (μg/mL)
**NtSt01**	500 250 125 62.50 31.25	80.03 ± 2.11 ** 75.52 ± 0.96 ** 71.63 ± 0.92 ** 67.63 ± 2.51 ** 62.35 ± 1.78 **	4.17
**NtSt02**	500 250 125 62.50 31.25	74.99 ± 1.53 *** 64.32 ± 1.85 *** 60.04 ± 0.86 *** 53.10 ± 2.05 *** 46.84 ± 0.67 ***	46.73
**Standard Drug**	500 250 125 62.50 31.25	91.01 ± 1.36 87.79 ± 1.27 82.33 ± 1.00 75.63 ± 0.86 71.07 ± 1.82	1.96

All the values are expressed as mean ± SEM compared with the standard. Two-way ANOVA followed by Dunnett’s test was applied. ** = *p* < 0.01, *** = *p* < 0.001 compared with standard drug.

**Table 3 plants-12-03591-t003:** Antioxidant potential of the phytosteroids.

Comp/Standard	Conc (μg/mL)	Percent Inhibition (Mean ± SEM)	IC_50_ (μg/mL)
**NtSt01**	500 250 125 62.50 31.25	61.05 ± 2.05 *** 55.79 ± 0.92 *** 47.08 ± 2.41 *** 42.53 ± 0.69 *** 37.25 ± 2.66 ***	142.76
**NtSt02**	500 250 125 62.50 31.25	59.12 ± 1.96 *** 52.01 ± 1.07 *** 42.99 ± 2.70 *** 33.08 ± 1.17 *** 28.52 ± 2.63 ***	223.43
**Standard Drug**	500 250 125 62.50 31.25	91.52 ± 2.52 85.04 ± 0.63 81.54 ± 1.28 77.87 ± 1.49 75.37 ± 1.69	0.74

All the values are expressed as mean ± SEM compared with the standard. Two-way ANOVA followed by Dunnett’s test was applied. *** = *p* < 0.001 compared with the standard drug.

**Table 4 plants-12-03591-t004:** Observational changes in blood glucose level in alloxan-induced fasting diabetic rats in mg/dL.

Groups	Day 1	Day 7	Day 14	Day 21	Day 28
**Normal control**	102 ± 2.36	106 ± 0.98	105 ± 1.77	110 ± 0.91	109 ± 1.07
**Diabetic control**	521 ± 1.28	517 ± 2.06	526 ± 1.00	524 ± 2.66	529 ± 2.80
**Glibenclamide (0.5 mg/kg)**	517 ± 0.77	398 ± 2.08	302 ± 1.38	207 ± 1.91	103 ± 1.70
**NtSt01 (1.0 mg/kg)**	519 ± 3.98	413 ± 1.87	325 ± 1.62	219 ± 2.87	116 ± 1.33

**Table 5 plants-12-03591-t005:** The effect of NtSt01 on body weight in grams of fasting rats.

Groups	Day 1	Day 7	Day 14	Day 21	Day 28
**Normal control**	226 ± 0.98	227 ± 1.27	228 ± 0.63	229 ± 1.67	233 ± 0.48
**Diabetic control**	232 ± 2.69	229 ± 1.98	225 ± 1.39	217 ± 2.69	214 ± 3.09
**Glibenclamide (0.5 mg/kg)**	235 ± 1.36	236 ± 1.37	239 ± 0.67	244 ± 2.38	249 ± 3.18
**NtSt01 (1.0 mg/kg)**	238 ± 0.33	240 ± 0.49	242 ± 1.04	245 ± 1.11	246 ± 0.11

**Table 6 plants-12-03591-t006:** The effect of NtSt01 on vital organ weights in grams of fasting rats.

Groups	Liver	Kidney	Pancreas	Heart
**Normal control**	9.16 ± 0.45	0.97 ± 0.13	0.88 ± 0.02	1.02 ± 0.11
**Diabetic control**	6.38 ± 0.10	1.26 ± 0.37	0.71 ± 0.13	0.83 ± 0.08
**Glibenclamide (0.5 mg/kg)**	9.21 ± 0.66	0.99 ± 0.16	0.84 ± 0.21	0.90 ± 0.02
**NtSt01 (1.0 mg/kg)**	9.83 ± 0.92	1.01 ± 0.15	0.92 ± 0.12	1.04 ± 0.13

**Table 7 plants-12-03591-t007:** Biochemical profile of blood in alloxan-induced diabetic rats.

Test	N.Control	D.Control	Standard	NtSt01 (1.0 mg/kg)	Unit	Reference Range
S.Creatinine	0.42 ± 0.02	0.96 ± 0.10	0.43 ± 0.01	0.6 ± 0.02	mg/dL	0.4–0.8
Blood Urea	19.1 ± 0.17	187 ± 3.65	20.2 ± 0.33	19.1 ± 2.36	mg/dL	15–22
S.bilurubin	0.71 ± 0.24	0.96 ± 0.01	0.81 ± 0.03	0.72 ± 0.77	mg/dL	Up to 1.0
SGPT(ALT)	29.4 ± 1.07	268 ± 1.37	37 ± 1.11	32 ± 0.25	U/L	17–30
S.ALK.Phosphatase	114.7 ± 0.39	159 ± 1.07	125 ± 1.22	134 ± 2.33	U/L	30–130

**Table 8 plants-12-03591-t008:** Antilipidemic effects of NtSt01 on alloxan-induced diabetic rats.

Groups	S.Cholesterol	S.Triglycerides	HDL	LDL
**Normal control**	52.06 ± 1.04	93.54 ± 1.37	43.50 ± 2.35	23.88 ± 0.82
**Diabetic control**	280.63 ± 2.33	456 ± 3.21	40.76 ± 1.49	165.5 ± 3.65
**Glibenclamide (0.5 mg/kg)**	59.98 ± 0.99	112 ± 0.99	32.55 ± 2.35	34.16 ± 2.66
**NtSt01 (1.0 mg/kg)**	77.45 ± 1.72	125 ± 3.65	38.10 ± 1.64	48.85 ± 1.08
**References Range**	10–54	26–145	Up to 50	10–54

## Data Availability

The data are presented in the manuscript and Appendix A.

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
