# Peer review of "Antidiabetic, Antihyperlipidemic, and Antioxidant Evaluation of Phytosteroids from *Notholirion thomsonianum* (Royle) Stapf"

_plants, 2023, doi:10.3390/plants12203591_

Round 1
Reviewer 1 Report
Dear authors,
I extend my congratulations on your paper titled "Antidiabetic, Anti-hyperlipidemic and Antioxidant Evaluations of Phytosteroids from Medicinal Plant Notholirion thomsonianum". Your manuscript is commendably organized. However, there are several significant aspects that require attention prior to publication. I would like to provide constructive feedback to enhance the quality of your work.
- Firstly, some words were poorly chosen, for example "destruction" (line 41), "for long time" (line 49), “vital organs” (line 92), “each time” (line 100); “very dangerous” (line 289); “slaughtered under the ethical procedure” (line 290); “good enough” (line 304). Additionally, a thorough proofreading is essential to correct typos, such as “5 Mm” (line 111); “were studies” (line 155). Seeking assistance from an expert in language editing is recommended.
- In the introduction (line 52) you write that there are two types of diabetes mellitus, which is not correct. Even if you don't give any more details, it's important to make it clear that there are different types of DM. It is therefore essential to correct this statement.
- Some parts of the text are written in a confusing way, for example, "The use of antidiabetic drugs has risk of number of adverse effects..." (line 57); “we have identified the phytosteroids within the hydroalcoholic extract” (line 88); “…assay by Brand William was used (line 129); “The food, water and light/dark cycles for animals were under the observation of ethical committee” (line 141); “On fasting mood the experimental animals were kept for 8-12 hours but…” (line 149; In fact the whole paragraph dealing with “Induction of diabetes” deserves to be better written!!); “as it highly sensitive reaction to temperature” (line 186); “Though both of our phytosteroids were antioxidant with low activity profiles, but it was still notable as a supplementary target” (line 253). It is imperative that you proofread and improve the text as a whole.
- In line 61, where you discuss the inhibition of enzymes like α-amylase and α-glucosidase, it would be valuable to briefly touch upon other potential pharmacological strategies as well.
- I cannot identify clearly what is your main objectives. To enhance the clarity of your objectives, ensure that readers are well-informed about the main goals of your study.
- For the DPPH assay, you need to improve the description. How and why was the absorbance adjusted to 0.75? Why did you choose the dilutions 31.25 to 500 ug/mL and how did you make these dilutions? Be more specific in your descriptions.
- The experimental design lacks a lot of important information. For how long did the treatments take place? How often was NtSt01 and glibenclamide administered? How were the animals euthanized?
- You often mention R1, R2, etc. in biochemical tests. That's not necessary! In fact, you must rewrite the sections 2.6.5, 2.6.6 and 2.6.7.
- What is the IUPAC name for NtSt01?
- The results in the 3.2 and 3.3 sections are described very confusingly. Consider review and rewrite. Besides, are the IC50 of 7.34 and 4.15 correct ?! They seem to be very, very low, as the same occurs for the “standard drug”. Verify the accuracy of the IC50 values
- Table 4 makes no sense at all. What is it supposed to show?
- You tested increasing concentrations of NtSt01 (from 200 to 2000 mg/ kg of weight), but in the end you used a dose of 1.0 mg/kg. Address the discrepancy between the tested concentrations of NtSt01 and the ultimately used dose of 1.0 mg/kg. Why and how did you choose this concentration?
- In Table 8 there is a “standard group” ?
- In line 318, one can read: “To control high glucose level in DM, the treatment of choice is therapy. Apart from this, α-amylase and α-glucosidase enzymes inhibitors are the important strategies for controlling hyperglycemia, as these enzymes convert starch into glucose ”. Come on guys, you can do better than this !!!!
“In OGT test”?!!! (line 338).
- The main criticism for publication of your manuscript comes now. Your discussion is merely a repetition of your results. That's not the way to do it! The discussion section needs to be improved a lot, a lot! Refine the discussion section to be more than a mere repetition of results and incorporate deeper insights, potential implications, and contextual comparisons, as well as the limitations of your study.
- As a tip: you use the term "vital organs" in several places. Academic texts need to be more precise, so consider describing which organs you are referring to. Besides, it sounds strange to think of any organ that isn't "vital", doesn't it?
Dear authors, while your work holds merit and deserves space in specialized literature, it necessitates thorough language review and substantial revisions in content. These suggestions are intended to support your progress, so please approach them with an open mind. Don't be discouraged by my comments, as they are designed to help you.
I wish you luck and that you continue to do good science.
It would be very useful for you to seek the help of an editing service that is committed to writing scientific articles.
Reviewer 2 Report
The main notes:
1. I suppose that the title of the article should not indicate "medicinal plant"- 'Antidiabetic, Anti-hyperlipidemic and Antioxidant Evaluations of Phytosteroids from medicinal plant Notholirion thomsonianum (Royle) Stapf (Liliaceae)'. It is due Notholirion thomsonianum is not included in any Pharmacopoeia of the world, and now it is only at the initial stage of studying its pharmacological properties. For instance, I found only 3 results for ' Notholirion thomsonianum' in the PubMed database - https://pubmed.ncbi.nlm.nih.gov/?term=Notholirion+thomsonianum&sort=date&size=20
2. According to the modern data of 'Plants of the World Online | Kew Science', it should be specifying the names of the scientists who described this species for science (after the Latin name of plant at the first mention) - Notholirion thomsonianum (Royle) Stapf.
3. The abbreviations GCMS, OGT, DM, TLC, etc. should be deciphered in the text during the first mention. The abbreviation RFTs is emerged only in line 182; thus, it could be removed. In general, the abbreviations should be carefully reviewed and sometimes corrected by the authors
4. The list of keywords should be widened to capture the main findings. For example, the 'in vitro', 'in vitro' could be added. The term 'phytochemistry' could be removed.
5. In Chapter 2, the authors should indicate what is the raw materials of the plant taken for research and where were it taken from (grown by themselves, harvested in nature or bought) as well as name and briefly describe the methods used to identify the compounds isolated from the plant. I suppose also that this sentence is incorrect ' We have collected this plant and have used in our previous research [29-31] – line 95.
6. The conditions of the chromatographic analysis are not indicated, and the corresponding chromatograms and spectral characteristics of the isolated compounds are not given. It should be added.
7. The moderate check is required for the English language and style through the text. Thus, the Abstract should be corrected, for instance: Diabetes mellitus (DM) is a metabolic complication and can be a big challenge to the human health. DM is the root cause of many life-threatening diseases. The natural products researchers are in continuous efforts to treat vital diseases in an economical and efficient way. In this research, we have exploited the phytosteroids from Notholirion thomsonianum for the treatment of DM. The structures of the phytosteroids NtSt01 and NtSt02 were confirmed with …… (GC-MS) and .... (NMR) analyses. In the in vitro α-glucosidase, α-amylase and DPPH assays, compound NtSt01 was found comparatively potent.....'
8. It is worth widening the botanical description of this plant in the Introduction. It is worth to be pointed, for instance, that 'The native range of this species is Afghanistan to Nepal. It is a bulbous geophyte [Plants of the World Online] https://powo.science.kew.org/taxon/urn:lsid:ipni.org:names:538786-1
9. The authors elaborated on too few new scientific articles published in the last 5 years (less than 1/3 of sources: 2023 - 2, 2022 – 5, 2021 – 3, 2020 -1, 2019 -3). Thus, the list of references could be renewed. At the same time, authors should reduce the level of self-citations. So, for example, for the name Sadiq A, there are more than 10 publications in the list of used sources as well as 9 for Mahnashi MH. In my opinion, the level of self-citation cannot be more than 10-15%.
10. The use of some terms should be made more universal, because they differ, for example, in lines 107 and 108: '2.3. Alpha Glucosidase Inhibition. The α-glucosidase inhibitory….' Similarly, the dimension that denotes "kilogram" is sometimes written with a capital letter (mg/Kg – lines 25, 162, 284, etc.), and in Tables 5-9 with a small letter (I believe that this is more correct ' mg/kg'). It is also more common to write milliliters as " mL" instead of "ml"ю
11. The Conclusions can be shortened and modified a little since the first two sentences do not reflect the generalized data from the authors' research "Diabetes mellitus is a metabolic complication and can be a big challenge to the human health. The herbal medicine and medicinally important compounds are considered safe drugs as compared to the synthetic. In an attempt to combat the diabetes and its consequences, we have used phytosteroid NtSt01 as a natural drug isolated from Notholirion thomsonianum..."
12. The italic type should be used everywhere for writing Latin names of species and for terms in vitro, in vivo - Lines 23, 87, 251, 454, etc.
The moderate check is required for the English language and style through the text.
Reviewer 3 Report
In this manuscript, the authors isolated two steroid based natural products from medicinal plant Notholirion thomsonium, characterised and evaluated their antidiabetic, antilipidemic and antioxidant properties.
One of the derivatives possess as good inhibitors as compared to the reference standards.
My comments are as follows.
1. The abstract and the title of the manuscript are presented in line with the objective of the work. The whole manuscript is presented with good scientific language and typographical errors are found to be minimal.
2. The importance and quality of the work is high due to the following reasons:
a) The authors have isolated the potent molecules and from natural sources and studies against multiple inhibition phenomenon for T2DM and results are found to be good.
b) The inhibition properties of one derivative are showing positive results and which is the very interesting.
3. The authors might incorporate schematic representation of all types of inhibition mechanism like alpha glucosidase, alpha amylase etc.
4. The molecules are isolated with standard synthetic protocols.
How much compound was isolated (in g or mg?)
Appearance of the compounds?
It is advisable to submit NMR data and spectra for each.
Absolute stereochemistry should be defined for both NtS01 and NtS02.
GCMS analysis is seems to be Ok. HRMS analysis is recommended.
The purity profile of the compounds should be provided.
5. All biological evaluation studies are performed using standard protocols.
The compound NtS01 seems to be good inhibitor and compatible to reference standard with less toxicity. For mechanistic analysis molecular docking studies recommended.
6. It is suggested to mention the importance of synthetic efforts towards development of antidiabetic medicine. In this regard, it is recommended to emphasis the importance of iminosugars and sugar derivatives as an antidiabetic agent and suggested to cite following relevant articles in the introduction section; i) https://doi.org/10.1002/anie.202217809 ii) Compain, P.; Martin, O. R. Iminosugars: From synthesis to therapeutic applications; Wiley-VCH : New York, 2007; pp 187−298 and iii) https://doi.org/10.24820/ark.5550190.p011.809.
Round 2
Reviewer 1 Report
Dear authors,
I am pleased to note that you have made several changes to the manuscript, which have undoubtedly contributed to its improvement. Nevertheless, there are a few remaining issues that warrant further clarification:
- Line 128: "We have collected this plant..." - Please specify the plant and which part(s) of the plant were collected.
- Line 132: "Afterwards, each time the solvent polarity was gradually increased by adding 5% ethyl acetate and the fractions were collected." - This sentence remains unclear. Please rephrase or provide additional context.
- Line 144: "600 μL phosphate buffer (0.1 M, pH 6.9) was added, and the mixture was at 37 °C incubated for 15 minutes." - Consider revising to: "600 μL of phosphate buffer (0.1 M, pH 6.9) was added, and the mixture was incubated at 37 °C for 15 minutes."
-Line 145: Clarify whether the concentration of p-nitrophenyl α-D-glucopyranoside is 5M or 5 mM.
-Line 204: Provide a precise description of how euthanasia was conducted, as not all readers may be familiar with the AVMA guidelines (version 2020).
- In section 2.6.7 (Liver function tests), it is little informative to know just the order and volumes of “Reagent 1”, “Reagent 2”, “R3”, “R4” were used. Instead of listing the order and volumes of reagents, it would be more informative just to cite the specific commercial kits used.
- Line 423: "This overall, results showed that the compound NtSt01 was effective to protect the vital organs from damage over the four weeks of experimental time." - Please remove the comma at the beginning of the sentence. Also, consider having other professors or researchers proofread your manuscript for typos and clarity.
- Line 461: "Natural products have minimum side effects to that of synthetic drugs, and hence researchers have focused plants to develop new drugs having minimum or no side effects." - The same as before, you can write a better sentence than that !
- Line 469: "In this study, an attempt was made to determine the antidiabetic potential of Notholirion thomsonianum through in-vitro (α-amylase and α-glucosidase) and in-vivo studies and it was revealed that our test samples have the potential to possess antidiabetic activity." - Simplify and clarify, for instance: “In this study we demonstrated the antidiabetic potential of Notholirion thomsonianum through in-vitro (α-amylase and α-glucosidase) and in-vivo studies”.
- Line 492: Once again, lets proofread what you wrote! “The vital organs were obtained after experiments completion and changes in weight of vital organs like liver, kidney, pancreases and heart were measured among all groups and no obvious changes were observed in the average weight of organs.”. What do you think about rewrite to something like this: “"No statistical differences in the weights of the liver, kidney, pancreas, and heart were observed among all groups." Is it not clearer, in your opinion?
- In this revised version I also think that the discussion still lacks depth. For example, you've provided some very interesting results on molecular docking studies, but you don't address these results at all in the discussion!
- In the discussion section, line 486 to 491, the result of the OGT is described. So far so good! But either I'm very mistaken or I haven't read about this methodology before. Besides, the RESULTS you found for this test should be in the RESULTS section!!!
- ONCE AGAIN the term "vital organs" appears in your conclusion. I urge you to revise the text as A WHOLE and avoid, as far as possible, being vague, not very direct, using expressions that are not objective.
Dear authors, I have recognized your potential for enhancing the manuscript, thus preparing it for a publication of excellence. Although it may appear taxing and disheartening, it remains imperative to undergo further revisions in both the manuscript's content and the depth of discussion regarding your results.
I hold the hope that you will consider and incorporate my suggestions, allowing me the anticipation of soon reading this work, as well as future contributions from your esteemed team in the specialized literature.
Best regards.
While not obligatory, and without compromising the publication's quality, it would greatly benefit the manuscript to undergo proofreading by an individual proficient in scientific English grammar.
Reviewer 2 Report
The main notes:
I suppose that the title of the article should not indicate "medicinal plant"- 'Antidiabetic, Anti-hyperlipidemic and Antioxidant Evaluations of Phytosteroids from medicinal plant Notholirion thomsonianum (Royle) Stapf '. It is due Notholirion thomsonianum is not included in any Pharmacopoeia of the world, and now it is only at the initial stage of studying its pharmacological properties
2. The list of keywords should be widened to capture the main findings. For example, the 'in vitro', and 'in vitro' could be added. The term 'phytochemistry' could be removed.
Minor editing of English language required
Reviewer 3 Report
The authors have addressed all my comments for this manuscript and answered the technical questions I have for this article. The article has been significantly improved after revising so and I recommend publication in Plants.
